# Essential Oil from *Glossogyne tenuifolia* Inhibits Lipopolysaccharide-Induced Inflammation-Associated Genes in Macro-Phage Cells via Suppression of NF-κB Signaling Pathway

**DOI:** 10.3390/plants12061241

**Published:** 2023-03-09

**Authors:** Wan-Teng Lin, Yen-Hua He, Yun-Hsin Lo, Yu-Ting Chiang, Sheng-Yang Wang, Ismail Bezirganoglu, K. J. Senthil Kumar

**Affiliations:** 1Department of Hospitality Management, College of Agriculture and Health, Tunghai University, Taichung 40704, Taiwan; 040770@thu.edu.tw (W.-T.L.);; 2Department of Forestry, National Chung Hsing University, Taichung 402, Taiwan; 3Department of Molecular Biology and Genetics, Erzurum Technical University, Erzurum-25050, Turkey; 4Bachelor Program of Biotechnology, National Chung Hsing University, Taichung 402, Taiwan

**Keywords:** *Glossogyne tenuifolia*, *Asteraceae*, essential oil, anti-inflammation, NF-κB

## Abstract

*Glossogyne tenuifolia* Cassini (*Hsiang-Ju* in Chinese) is a perennial herb native to Taiwan. It was used in traditional Chinese medicine (TCM) as an antipyretic, anti-inflammatory, and hepatoprotective agent. Recent studies have shown that extracts of *G. tenuifolia* possess various bioactivities, including anti-oxidant, anti-inflammatory, immunomodulation, and anti-cancer properties. However, the pharmacological activities of *G. tenuifolia* essential oils have not been studied. In this study, we extracted essential oil from air-dried *G. tenuifolia* plants, then investigated the anti-inflammatory potential of *G. tenuifolia* essential oil (GTEO) on lipopolysaccharide (LPS)-induced inflammation in murine macrophage cells (RAW 264.7) in vitro. Treatment with GTEO (25, 50, and 100 μg/mL) significantly as well as dose-dependently inhibited LPS-induced pro-inflammatory molecules, such as nitric oxide (NO) and prostaglandin E_2_ (PGE_2_) production, without causing cytotoxicity. Q-PCR and immunoblotting analysis revealed that the inhibition of NO and PGE_2_ was caused by downregulation of their corresponding mediator genes, inducible nitric oxide synthase (*iNOS*), and cyclooxygenase-2 (*COX-2*), respectively. Immunofluorescence and luciferase reporter assays revealed that the inhibition of *iNOS* and *COX-2* genes by GTEO was associated with the suppression of nuclear export and transcriptional activation of the redox-sensitive transcription factor, nuclear factor -κB (NF-κB). In addition, GTEO treatment significantly inhibited phosphorylation and proteosomal degradation of the inhibitor of NF-κB (I-κBα), an endogenous repressor of NF-κB. Moreover, treatment with GTEO significantly blocked the LPS-mediated activation of inhibitory κB kinase α (IKKα), an upstream kinase of the I-κBα. Furthermore, *p*-cymene, β-myrcene, β-cedrene, *cis*-β-ocimene, α-pinene, and _D_-limonene were represented as major components of GTEO. We found that treatment with *p*-cymene, α-pinene, and _D_-limonene were significantly inhibiting LPS-induced NO production in RAW 264.7 cells. Taken together, these results strongly suggest that GTEO inhibits inflammation through the downregulation of NF-κB-mediated inflammatory genes and pro-inflammatory molecules in macrophage cells.

## 1. Introduction

Inflammation is a complex and crucial physiological response to many pathological conditions, including microbial invasion and tissue injury, which is characterized by redness, swelling, and pain [1]. As a result of bacterial infection, an endotoxin (lipopolysaccharide, LPS) is produced by Gram-negative bacteria which leads to many pathological symptoms, such as pyrogenicity, increased circulating leukocytes, macrophage activation, anti-platelet aggregation, and increased capillary permeability [2]. Generally, local inflammatory responses are benign as long as the process is properly regulated to keep the cells and inflammatory mediators sequestered. Moreover, the uncontrolled inflammatory response can result in death when it is not controlled, as is the case with anaphylactic shock, as well as with chronic inflammatory diseases such as arthritis and gout [1].

As macrophages become activated by endotoxins or pathogenic microbes; they produce a variety of pro-inflammatory molecules such as nitric oxide (NO), prostaglandin E_2_ (PGE_2_), tumor necrosis factor-α (TNF-α), and interleukin-1β/6 (IL-1β/6) [3]. There are a variety of inflammatory disorders associated with the constant secretion of these molecules which include rheumatoid arthritis, pulmonary fibrosis, asthma, hepatitis, and hepatitis B [1]. Therefore, inhibiting these pro-inflammatory molecules is a promising strategy to minimize the burden of inflammatory diseases.

Traditional Chinese medicine (TCM) has been used to treat chronic diseases with food supplements and plant-based medicines. A number of TCMs contain essential oils (EOs), which are complex mixtures of volatile compounds obtained from plants through various extraction procedures, including azeotropic distillation (hydro-distillation, hydro-diffusion, and steam distillation) and extraction with solvents. A growing number of in vitro and in vivo studies have demonstrated that EOs possess various bioactivities, including antioxidant, anti-microbial, anti-inflammatory, and anti-cancer properties [4,5].

*Glossogyne tenuifolia*, locally known as *Hsiang-Ju,* is a perennial plant native to Penghu island, Taiwan, and also distributed in Southern Asia and Australia [6]. The herb has been used for making herbal tea on Penghu Island for centuries to treat pyrexia, hepatitis, and inflammation [7]. Recent studies have demonstrated that various extracts of *G. tenuifolia* possess a wide range of bioactivities, including anti-microbial [8], anti-inflammatory [9,10], anti-fatigue [11], antioxidant [12,13], antiviral [10], anti-angiogenesis [14], anti-cancer [15], hepatoprotection [16,17], and immunomodulatory effects [6,18]. In addition, Chyau, et al. [19] extracted essential oil from *G. tenuifolia* using simultaneous steam distillation and solvent extraction methods. A total of 62 compounds were isolated, including terpenes, which are the main constituents of *G. tenuifolia* essential oil. To the best of our knowledge, the bioactivities of *G. tenuifolia* essential oil (GTEO) have not been studied. Therefore, the present study was aimed at investigating the anti-inflammatory effects of GTEO on LPS-induced inflammation in murine macrophage cells (RAW 264.7) and delineating the underlying mechanism.

## 2. Results

### 2.1. Effect of GTEO on Cell Viability in LPS-Activated RAW 264.7 Cells

Prior to the in vitro anti-inflammatory assessment, the cytotoxicity of GTEO to the murine macrophages was determined. RAW 264.7 cells were incubated with increasing concentrations of GTEO (25–200 μg/mL) for 24 h. Next, the MTT colorimetric assay was used to measure the cell viability. As shown in Figure 1A, the number of RAW 264.7 cells were significantly increased following stimulation with LPS (1 μg/mL), whereas cell viability was not affected by GTEO doses up to 200 μg/mL, indicating that within the concentration ranges tested, GTEO was not cytotoxic to RAW 264.7 cells.

### 2.2. GTEO Inhibits LPS-Induced NO and PGE_2_ Production in RAW 264.7 Cells

In this study, a significant increase in NO production (39.58 ± 2.31 μM) from 3.1 ± 0.28 μM (control) was observed upon stimulation with LPS, whereas co-treatment with GTEO significantly and dose-dependently reduced LPS-stimulated NO production to 32.37 ± 12.67 μM, 20.54 ± 2.3 μM, and 10.0 ± 0.74 μM by 25, 50, and 100 μg/mL, respectively (Figure 1B). In the next step, we examined the level of PGE_2_ in culture media. When cells were incubated with LPS for 24 h, PGE_2_ production markedly increased from 54 ± 13 pg/mL to 666 ± 47 pg/mL. This increase was significantly as well as dose-dependently inhibited by GTEO, as evidenced by the PGE_2_ production, which was reduced to 524 ± 41, 399 ± 40, and 153 ± 7 pg/mL by 25, 50, and 100 μg/mL GTEO, respectively (Figure 1C).

### 2.3. GTEO Inhibits LPS-Induced iNOS and COX-2 Expression in RAW 264.7 Cells

In order to determine whether the inhibitory effects of GTEO on the pro-inflammatory molecules (NO and PGE_2_) are related to the modulation of their corresponding mediators, *iNOS* and *COX-2* genes and protein expression levels were examined. As shown in Figure 2A,B, strong *iNOS* and *COX-2* mRNA expression levels were observed upon stimulation with LPS. Indeed, treatment with GTEO significantly down-regulated LPS-induced *iNOS* and *COX-2* mRNA expression in a dose-dependent manner. To further confirm this effect at the translational level, immunoblotting was performed. As shown in Figure 2C, LPS-stimulation markedly increased iNOS and COX-2 protein expression, while iNOS and COX-2 levels were low or undetectable in unstimulated RAW 264.7 cells. Indeed, co-treatment with GTEO significantly and dose-dependently inhibited iNOS and COX-2 protein levels in LPS-stimulated RAW 264.7 cells.

### 2.4. GTEO Suppressed LPS-Induced NF-κB Transcriptional Activity in RAW 264.7 Cells

NF-κB is a family of inducible transcription factors that are involved in a wide array of inflammatory processes by up-regulating several inflammation-associated genes, including *iNOS*, *COX-2*, *TNF-α*, *IL-1β,* etc [20]. In order to determine whether the GTEO-mediated downregulation of *iNOS* and *COX-2* genes was caused by the suppression of NF-κB activity, a luciferase reporter assay was performed. As shown in Figure 3A, regarding treatment with LPS a 8.6-fold increase in NF-κB reporter activity has been observed, whereas co-treatment with GTEO significantly as well as dose-dependently suppressed NF-κB reporter activity, as evidenced by the fact that reporter activity was reduced to 6.0, 4.4, and 2.4-fold by 25, 50, and 100 μg/mL, respectively. The nuclear export of cytoplasmic NF-κB is another hallmark involved in NF-κB transcriptional activity; immunofluorescence analysis enumerated LPS-stimulation markedly increased NF-κB nuclear translocation, while a significant reduction in NF-κB level in the nucleus was observed in cells which were co-treated with GTEO (Figure 3B). These results suggesting that GTEO might interfere with the dissociation of I-κB from the NF-κB/I-κB cytosolic complex, hence inhibiting the nuclear translocation of NF-κB.

The process of transcriptional activation followed by nuclear translocation of NF-κB is regulated by the phosphorylation and proteosomal degradation of its endogenous repressor, I-κBα [21]. Therefore, the effect of GTEO on total and phosphorylated I-κBα protein expression was determined in LPS-induced RAW 264.7 cells. Western blot analysis exhibited a remarkable increase in the phosphorylation of I-κBα at Ser32/36 after LPS stimulation, while co-treatment with GTEO showed a significant reduction in LPS-induced phosphorylation of I-κBα which corresponded directly to the significant increase in cytosolic I-κBα (Figure 4). In addition, the phosphorylation of IKKα, an up-stream kinase of I-κBα, was significantly inhibited by GTEO in a dose-dependent manner (Figure 4). However, neither LPS nor GTEO affected the total IKKα level.

### 2.5. Chemical Compositions of G. tenuifolia Essential Oil

The yield of *G. tenuifolia* essential oil (GTEO) obtained by hydrodistillation was 0.08% (*w*/*w*). The major chemical constituents of the essential oil and their relative amounts were determined by GC–MS analysis. The GC–MS profiles and the major compounds of GTEO are shown in Appendix A. The relative contents (%) in GTEO are shown in Table 1. A total of 21 compounds were identified in GTEO, accounting for 95.95% of the whole oil. The major components in GTEO were *p*-cymene (35.5%), β-myrcene (14.68), β-cedrene (9.8), *cis*-β-ocimene (8.49), α-pinene (6.69), and _D_-limonene (5.17), which made up around 74.96% of the content of the GTEO.

### 2.6. Nitric Oxide Inhibitory Effects of Major Constituents of GTEO

A previous study reported that α-pinene, β-pinene, limonene, *p*-cymene, and β-phellandrene are major components of GTEO [19]. In contrast, β-phallandrene was not found, while *cis*-β-ocimene and β-cedrene were found in our sample. In order to further investigate GTEO’s anti-inflammatory effects, its major compounds were tested for their NO inhibitory effects. Prior to investigating anti-inflammatory effects, MTT tests were performed to assess cytotoxicity. Treatment with either *p*-cymene, α-pinene, β-pinene, or limonene did not display cytotoxicity in RAW 264.7 cells up to a concentration of 100 μM, while *cis*-β-ocimene and β-myrcene exhibited cytotoxicity over a dose of 25 μM for 24 h (Appendix A). Therefore, the NO inhibitory effects of p-cymene, α-pinene, β-pinene, and D-limonene were investigated. Interestingly, a significant and dose-dependent inhibition of NO production was achieved by p-cymene, α-pinene, and limonene in LPS-stimulated RAW 264.7 cells. However, treatment with β-pinene failed to modulate LPS-induced NO production (Figure 5). These results clearly indicate that α-pinene, limonene, and *p*-cymene could be responsible for the anti-inflammatory effects of GTEO.

## 3. Discussion

Compared to conventional therapies, complementary and alternative treatments have been increasingly popular as a growing body of evidence has clarified their efficacy, safety, and mechanism of action. The herb *G. tenuifolia* has been used for thousands of years to treat inflammation and liver diseases in Taiwan [16]. A number of recent studies have revealed that *G. tenuifolis* has various bio-pharmaceutical properties that extend beyond its original use. Accordingly, *G. tenuifolia* possesses anti-microbial [8], anti-inflammatory [9,10], anti-fatigue [11], antioxidant [12,13], antiviral [10], anti-angiogenesis [14], anti-cancer [15], hepatoprotection [16,17], and immunomodulatory effects [6,18]. In particular, *G. tenuifolia* has been shown to have powerful anti-inflammatory properties. Wu et al. [9] reported that ethanol extract of *G. tenuifolia* inhibits LPS-induced NO, PGE_2_, TNF-α, IL-1β, IL-6, and IL-12 production in cultured RAW 264.7 cells. The inhibition of NO and PGE_2_ was reasoned by decrease in the protein and mRNA levels of their mediators iNOS and COX-2, respectively. Further analysis revealed that *G. tenuifolia* attenuates inflammatory mediators in LPS-induced macrophages by suppressing the transcriptional activity of NF-κB. A following study [10] also reported that ethanol extract of *G. tenuifolia* inhibits LPS-induced TNF-α and IL-6 production in human whole blood and peripheral blood mononuclear cells (PBMC), and secretion of IFN-γ in PHA-stimulated human whole blood. In addition, this study also revealed that ethanol extract of *G. tenuifolia* had potent anti-hepatitis B virus (HBV) activity on the human hepatocellular carcinoma cell line (PLC/PRF/5). Another study further explained *G. tenuifolia*’s suppressive effect on NF-κB transcriptional activity in TNF-α-induced human umbilical vein endothelial cells (HUVECs) [22]. Ethanol extract of *G. tenuifolia* and its major components, luteolin and luteolin-7-glucoside, inhibits adhesion molecules including intercellular adhesion molecule-1 (ICAM-1) and vascular cell adhesion molecule-1 (VCAM-1), which in turn are mediated through blocking the activation and nuclear translocation of NF-κB [22]. In contrast, ethanol, *n*-hexane, ethyl acetate, and methanol extracts of the whole, areal, and roots of *G. tenuifolia* increased NO production in HUVECs; this increase was associated with the upregulation of endothelial nitric oxide synthase (eNOS) [23]. Several studies have investigated the anti-inflammatory mechanism of *G. tenuifolia*, but all of them have focused on various extracts of *G. tenuifolia*. However, the bioactivities, including the anti-inflammatory property of *G. tenuifolia* essential oil, are largely unexplored. In this study, it was found that GTEO treatment inhibited pro-inflammatory molecules in LPS-induced macrophages by suppressing NF-κB signaling.

It has been well demonstrated that macrophages are critical immune effector cells and that they are sensitive to bacterial endotoxins such as LPS. In response to LPS stimulation, macrophages release pro-inflammatory cytokines, chemokines, and adhesion molecules [24]. NO, a prominent pro-inflammatory chemokine, functions as an intracellular messenger which regulates vasodilation and removes pathogens and tumor cells. However, aberrant production of NO can lead to several pathological conditions, including inflammation, asthma, diabetes, and septic shock [25]. Stimulating RAW 264.7 cells with LPS, NO is secreted into the culture media, which is measured by nitrite, a non-volatile breakdown product [26]. Our findings indicate that GTEO significantly inhibits LPS-induced NO production in RAW 264.7 cells. It is known that activated macrophages secrete PGE_2_ as one of their stable prostanoids, which can be detected in the culture media [27]. We found that GTEO strongly inhibited LPS-induced PGE_2_ secretion in RAW 264.7 cells. iNOS is an enzyme involved in reactive oxygen and nitrogen metabolism, while COX-2 catalyzes the rate-limiting step in prostaglandin biosynthesis [28]. Indeed, NO derived from iNOS and PGE_2_ is synthesized by COX-2. Therefore, we hypothesized that the inhibition of NO and PGE_2_ may result in downregulation of their mediator genes—*iNOS* and *COX-2*. As we expected, GTEO significantly and dose-dependently down-regulated iNOS and COX-2 expression at both the transcriptional and translational levels.

The endotoxin LPS can activate NF-κB, a redox-sensitive transcription factor, and leads to the transcriptional activation of response genes such as *iNOS* and *COX-2* [20]. Therefore, we determined whether suppression of NF-κB activity was a reason for iNOS and COX-2 downregulation. GTEO inhibits NF-κB activity, as evidenced by reduced promoter activity followed by nuclear export determined by a luciferase reporter assay and immunofluorescence, respectively. The nuclear translocation of the p65/p50 complex is an essential step involved in NF-κB transcriptional activation, which is controlled by phosphorylation and degradation of its repressor, I-κB [20]. In this study, we found that LPS treatment dramatically increased phosphorylation of I-κBα at Ser32/36 residues, whereas GTEO treatment significantly reduced I-κBα phosphorylation and restored the total level, thereby inhibiting NF-κB activity. Additionally, I-κBα phosphorylation is regulated by its upstream kinase IKKα [20]. Upon stimulation by endotoxin, IKKα is phosphorylated and further activates I-κBα. Our data indicate that GTEO treatment significantly inhibited LPS-mediated IKKα phorphorylation in a dose-dependent manner. However, the total form of IKKα was unaffected by either LPS or GTEO.

Chyau et al. [19] extracted *G. tenuifolia* essential oil from dried herbs, which are collected in the four seasons in Taiwan. Chemical fingerprint analysis revealed that essential oils from the four seasons had similar volatile profiles. In total, 62 compounds were isolated using the simultaneous steam distillation and solvent extraction method. Among them, 30 compounds were chemically identified, including 13 terpenes, 16 oxygen-containing compounds, and one other compound. Indeed, terpenes constituted 61.3–76.0% of the total amount, with 69.1% on average. Eight compounds, *p*-cymene, β-pinene, β-phellandrene, limonene, cryptone, α-pinene, 4-terpineol, and γ-muurolene were identified as the most abundant compounds, making up 71.5% of the average GTEO.

Several studies have investigated the anti-inflammatory effects of terpenes isolated from various essential oils. Kim et al. [29] reported that α-pinene inhibits the LPS-induced secretion of NO, TNF-α, and IL-6 in murine peritoneal macrophages through suppression of mitogen-activated protein kinases (MAPKs) and NF-κB signaling pathways. A study conducted on the anti-inflammatory property of β-pinene demonstrated that treatment with β-pinene inhibited carrageenan-induced paw edema in diabetic mice [30]. Limonene, a mono-terpene has been reported to inhibits LPS-induced NO and PGE_2_ production in RAW 264.7 cells via down-regulating their corresponding mediators, iNOS and COX-2 [31]. *p*-Cymene is a monocyclic monoterpene reported to inhibit LPS-induced NO and TNF-α production in murine peritoneal macrophages [32]. Likewise, 4-terpineol, the main component of tea tree essential oil, inhibits LPS-induced TNF-α, IL-1β, IL-8, IL-10, and PGE_2_ in human peripheral blood monocytes [33]. In the present study, we also found that α-pinene, limonene, and *p*-cymene significantly inhibited LPS-induced NO production in RAW 264.7 cells. According to our result, *cis*-β-ocimene and β-myrcene did not show NO inhibitory effects in the tested conditions up to a dose of 100 μM. However, both of these compounds exhibited cytotoxicity over a dose of 25 μM; thus, it may be of interest to further investigate its cytotoxic effects in various cancer cells.

## 4. Materials and Methods

### 4.1. Preparation of G. tenuifolia Essential Oil

*G. tenuifolia* was collected in March 2022 from the Penghu Islands, Taiwan. 250 g of the air-dried and chopped whole plant of *G. tenuifolia* and 1.5 L of distilled water were added into 2 L flask (Clevenger-type apparatus), and hydrodistillation was carried out for 6 h. On completion of the extraction process, essential oil was collected. The oil yield was calculated as: Yield (%) = (weight of essential oil recovered/weight of air-dried whole plant) × 100. The oil content was then determined. *G. tenuifolia* essential oil (GTEO) was stored in airtight sample vials prior to bioactivity evaluation.

### 4.2. Chemicals and Reagents

Dulbecco’s Modified Eagle’s medium (DMEM), fetal bovine serum (FBS), glutamine, and penicillin/streptomycin were obtained from Life Technologies (Grand Island, NY, USA). 3-(4,5-dimethyl-thiazol-2-yl)-2,5-diphenyl tetrazolium bromide (MTT) and 4′-6-diamidino-2-phenylindole (DAPI) were obtained from Sigma-Aldrich (St. Louis, MO, USA). Specific antibodies against iNOS, COX-2, IKKα, phos-IKKα, I-κBα, phos-I-κBα, NF-κB, and horseradish peroxidase (HRP)-conjugated anti-goat, anti-rabbit, and anti-mouse IgG secondary antibodies were obtained from Cell Signaling Technology (Danvers, MA, USA). Antibodies against glyceraldehyde 3-phosphate dehydrogenase (GAPDH) was obtained from Santa Cruz Biotechnology Inc. (Dallas, TX, USA). All other chemicals were of the highest grade commercially available and supplied either by Merck (Darmstadt, Germany) or Sigma (St. Louis, MO, USA).

### 4.3. Cell Culture and Cell Viability Assay

RAW 264.7 murine macrophage cells were obtained from the American Type Culture Collection (ATCC, Manassas, VA, USA). As recommended by ATCC, cells were cultured at 37 °C in DMEM with 10% FBS, 4.5 g/L glucose, 4.5 mM glutamine, 100 units/mL penicillin, and 100 g/mL streptomycin. An MTT colorimetric assay was used to determine cell viability. Briefly, in 96-well culture plates, RAW 264.7 cells (1 × 10^4^ cells) were seeded. We incubated cells with varying doses of GTEO (25–200 µg/mL) in the presence or absence of LPS (1 μg/mL) for 24 h. After treatment, the culture medium was removed and cells were incubated with MTT (10 µg/mL) in 200 µL of fresh DMEM for 1 h at 37 °C. Violet formazan crystals generated by MTT were dissolved in 200 μL of DMSO, and an ELISA microplate reader was used to measure absorbance at 570 nm (A_570_) and the background control at 630 nm (A_630_) (µQuant, Bio-Tek Instruments, Winooski, VA, USA). Cell viability (%) was calculated as: [(A_570_ − A_630_ of treated cells/(A_570_ − A_630_) of untreated cells) × 100.

### 4.4. Determination of NO and PGE_2_

NO levels in the culture media were determined using the Greiss reaction assay as described previously [34]. Briefly, RAW 264.7 cells (1 × 10^4^ cells/well) were seeded in a 96-well culture plate. Cells were treated with LPS in the presence or absence of various doses of GTEO (25–100 μg/mL) for 24 h. After treatment, an equal volume of culture supernatants was mixed with Griess reagent and incubated at room temperature for 30 min. The intercellular level of nitrate, a major stable product of NO, was measured with an ELISA microplate reader at 540 nm (µQuant, Bio-Tek Instruments, Winooski, VA, USA). On the other hand, intercellular PGE_2_ levels were determined using an EIA kit (R&D Systems, Minneapolis, MN, USA) according to the manufacturer’s protocols.

### 4.5. RNA Extraction and Q-PCR Analyses

RAW 264.7 cells (1 × 10^6^ cells/dish in 6-cm dish) were treated with LPS in the presence or absence of various doses of GTEO (25–100 μg/mL) for 6 h. After treatment, total RNA was extracted from cultured RAW 264.7 cells using the Trizol Reagent (Thermo Fisher Scientific, Waltham, MA, USA). A NanoVue Plus spectrophotometer (GE Health Care Life Sciences, Chicago, IL, USA) was used to quantify total RNA concentration. Real-time PCR detection system and software (Applied Biosystems, Foster City, CA, USA) was used for quantitative PCR. A SupeScript IV reverse transcriptase kit (Invitrogen, Waltham, MA, USA) was used to generate first-strand cDNA. A qPCR reaction was performed to quantify mRNA expression for genes of interest using equal volumes of cDNA, forward and reverse primers (10 μM), and power SYBR Green Master Mix (Applied Biosystems) under the following conditions: 96 °C for 3 min followed by 40 cycles at 96 °C for 1 min, 50 °C for 30 s, and 72 °C for 90 s. The primer sequences of each gene for qPCR were as follows. iNOS: forward primer (F), 5′-TCCTACACCACACCAAAC-3′; reverse primer (R), 5′-CTCCAATCTCTGCCTATCC-3″ COX-2: forward primer (F), 5′-CCTCTGCGATGCTCTTCC-3′; reverse primer (R), 5′-TCACACTTATACTGGTCAAATCC-3′; GAPDH: forward primer (F), 5′-TCAACGGCACAGTCAAGG-3′; reverse (R), 5′-ACTCCACGACATACTCAGC-3′. The copy number of each transcript was calculated as the relative copy number normalized by the GAPDH copy number. The relative abundance of target mRNA for each sample was calculated from the ^Δ^Ct values for the target and endogenous reference gene GAPDH using the ^2Δ^Ct cycle threshold method.

### 4.6. Protein Extraction and Immunoblotting

RAW264.7 cells (1 × 10^6^ cells/dish in 6-cm dish) were co-treated with LPS with or without various doses of GTEO (25–100 μg/mL) for various time points. After treatment, cells were detached and washed in cold PBS twice. Then, they were lysed in a radioimmunoprecipitation assay (RIPA) buffer (Pierce Biotechnology, Rockford, IL, USA). The concentrations of proteins were determined using a Bio-Rad protein assay reagent (Bio-Rad Laboratories, Hercules, CA, USA). Equal amounts of protein samples (60–100 mg/well) along with sample dye were denatured for 5 min at 94 °C. SDS-PAGE was used to separate the protein samples, followed by overnight transfer onto polyvinylidene fluoride (PVDF) membranes. The membranes were blocked with 0.1% Tween-20 in PBS containing 5% non-fat skim milk for 30 min at room temperature, reacted with primary antibodies for 2 h, and then incubated with either HRP-conjugated goat anti-rabbit or anti-mouse antibodies for 1 h. An enhanced chemiluminescence reagent (Advansta, Inc., San Jose, CA, USA) was used to develop the immunoblots, images were captured by the ChemiDoc XRS^+^ docking system, and the protein bands were quantified by using Imagelab software (Bio-Rad Laboratories, Hercules, CA, USA).

### 4.7. Immunofluorescence Assay

RAW 264.7 cells at a density of 1 × 10^4^ cells/well were cultured in an 8-well glass Nunc Lab-Tek chamber slide (Thermo Fisher Scientific) treated with LPS in the presence or absence of various doses of GTEO (25–100 μg/mL) for 1 h. After treatment, cells were washed with PBS twice, fixed in 4% paraformaldehyde in PBS for 15 min at room temperature, permeabilized with Triton X-100 in PBS for 10 min, washed and blocked with 10% FBS in PBS, and then incubated for 2 h with anti-NF-κB antibody in 1.5% FBS. The cells were then incubated with the fluorescein isothiocyanate (FITC) and conjugated with a secondary antibody for another 1 h in 6% BSA. Next, the nucleus was stained with 1 μg/mL DAPI for 5 min, washed with PBS, and visualized using a florescence microscope (Motic Electronic Group, Fujian, China) at 100× magnification.

### 4.8. Luciferase Reporter Assay

A dual-luciferase reporter assay system (Promega, Madison, WI, USA) was used to measure NF-κB transcriptional activity as described previously [34]. Briefly, RAW 264.7 cells were incubated for 5 h in DMEM without antibiotics in a 24-well culture plate that had reached 70–80% confluency. By using lipofectamine 2000 (Invitrogen), cells were transfected with a pcDNA vector or an NF-κB construct along with β-galactosidase. Cells were treated with LPS in the presence or absence of various doses of GTEO (25–100 μg/mL) for 2 h. The relative fluorescence intensity was quantified using a spectrometer (Hidex Oy, Turku, Finland) at 405 nm (A_405_). The fold increase of luminescence activity was calculated as (A_405_ of treated cells/A_405_ of untreated cells). The luciferase activity was normalized to β-galactosidase activity in the cell lysates.

### 4.9. GC-MS Analysis

To determine the chemical composition of GTEO, GC-MS analysis was carried out using an ITQ 900 mass spectrometer coupled with a DB-5MS column as described previously [35]. The temperature program was as follows: 45 °C for 3 min, then increased to 3 °C/min to 180 °C, and then increased to 10 °C/min to 280 °C hold for 5 min. The other parameters were injection temperature, 240 °C; ion source temperature, 200 °C; EI, 70 eV; carrier gas, He 1 mL/min; and mass scan range, 40–600 *m*/*z*. The volatile compounds were identified by the Wiley/NBS Registry of mass spectral databases (V. 8.0, Hoboken, NJ, USA), National Institute of Standards and Technology (NIST) Ver. 2.0 GC/MS libraries, and the Kovats indices were calculated for all volatile constituents using a homologous series of *n*-alkanes C_9_–C_24_ [36]. The major components were identified by co-injection with standards (wherever possible).

### 4.10. Statistical Analysis

Data are expressed as mean ± SD. All data were analyzed using the statistical software Graphpad Prism version 6.0 for Windows (GraphPad Software, San Diego, CA, USA). Statistical analysis was performed using one-way ANOVA followed by Dunnett’s test for multiple comparisons. A *p*-value of less than 0.001 ^Δ^ was considered statistically significant for the LPS alone vs. control group. *p*-values of less than 0.05 *, 0.01 **, and 0.001 *** were considered statistically significant for the LPS + GTEO treatment groups vs. LPS alone treatment group.

## 5. Conclusions

In the present study, we report for the first time the anti-inflammatory properties of *G. tenuifolia* essential oil as well as its mechanism of action. This study demonstrates that GTEO inhibits the production of the proinflammatory molecules NO and PGE_2_ by downregulating their corresponding mediators, iNOS and COX-2, in LPS-induced RAW 264.7 macrophage cells without attribution to cytotoxicity as assessed by the MTT assay. The inhibitory effect was attributed to the suppression of NF-κB transcriptional activation, since NF-κB is one of the key transcription factors responsible for regulating inflammation-related genes (Figure 6). In addition, the major compounds of GTEO, including α-pinene, limonene, and *p*-cymene inhibited LPS-induced NO production. Taken together, the inhibition of pro-inflammatory molecules via NF-κB by GTEO would be a possible therapeutic approach to the treatment of inflammation-associated genes.

## Figures and Tables

**Figure 1 plants-12-01241-f001:**
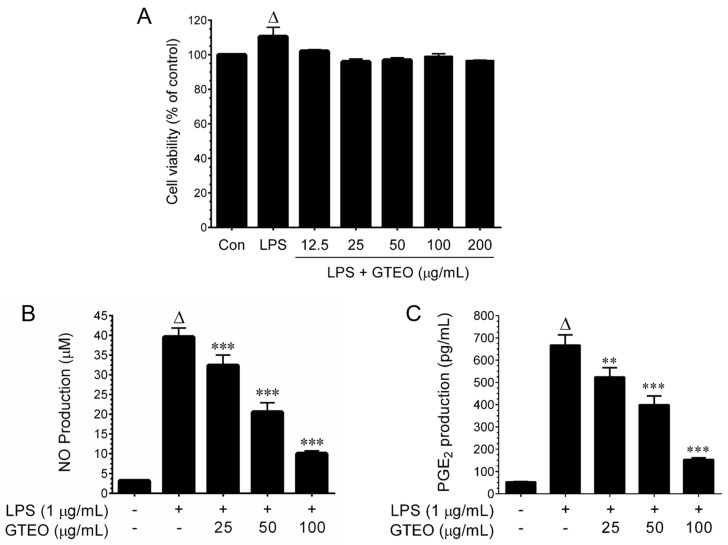
Effect of GTEO on cell viability and LPS-induced NO and PGE_2_ production. (**A**) RAW 264.7 cells were stimulated with LPS (1 μg/mL) and incubated in the presence or absence of increasing concentrations (25–200 µg/mL) of GTEO for 24 h. The cell viability was determined by the MTT colorimetric assay. (**B**) The nitrite concentration in the culture media was determined by the Griess reagent assay. (**C**) PGE_2_ levels in the culture media were measured by a commercially available assay kit as described in materials and methods. Data are reported as mean ± SD of three independent experiments. ^Δ^
*p* < 0.001 indicates a significant difference between the control and LPS-only treated groups. ** *p* < 0.01, and *** *p* < 0.001 show significant differences between the LPS-alone and GTEO treatment groups.

**Figure 2 plants-12-01241-f002:**
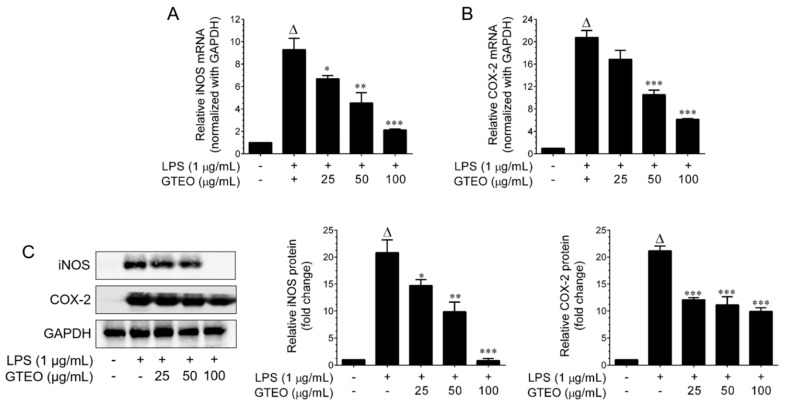
Effect of GTEO on LPS-induced iNOS and COX-2 expression. (**A**,**B**) RAW 264.7 cells were stimulated with LPS (1 μg/mL) and incubated in the presence or absence of increasing concentrations (25–200 µg/mL) of GTEO for 6 h. Total RNA was extracted and the transcription levels of *iNOS* and *COX-2* were quantified by Q-PCR. The Δ^ct^ values of *iNOS* and *COX-2* mRNAs were normalized with *GAPDH* mRNA. (**C**) RAW 264.7 cells were co-incubated with various doses of GTEO (25, 50, and 100 μg/mL) and 1 μg/mL LPS for 24 h. Protein expression levels of iNOS and COX-2 were determined by immunoblotting with specific antibodies. GAPDH served as an internal loading control. Histogram shows the relative protein expression levels of iNOS and COX-2, which are normalized with an internal control GAPDH. Data are reported as mean ± SD of three independent experiments. ^Δ^
*p* < 0.001 indicates a significant difference between the control and LPS-only treated groups. * *p* < 0.05, ** *p* < 0.01, and *** *p* < 0.001 show significant differences between the LPS-alone and GTEO treatment group.

**Figure 3 plants-12-01241-f003:**
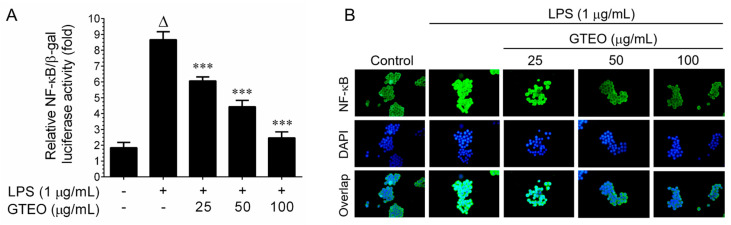
Effect of GTEO on LPS-induced transcriptional activation of NF-κB. (**A**) RAW 264. 7 cells were co-transfected with NF-κB harboring luciferase reporter construct. After transfection, cells were stimulated with LPS (1 μg/mL) and incubated in the presence or absence of GTEO (25–100 μg/mL) and in the presence of 1 μg/mL LPS for 2 h. Luciferase activity was determined and normalized with β-galactose activity. (**B**) The LPS-induced nuclear export of NF-κB was determined by immunofluorescence staining as described in materials and methods. Data are reported as mean ± SD of three independent experiments. ^Δ^
*p* < 0.001 indicates a significant difference between the control and LPS-only treated groups. *** *p* < 0.001 show significant differences between the LPS-alone and GTEO treatment groups.

**Figure 4 plants-12-01241-f004:**
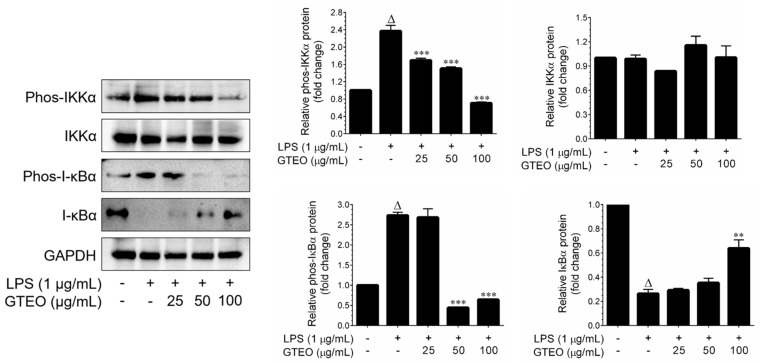
Effect of GTEO on LPS-induced I-κBα instability followed by activation of IKKα. RAW 264.7 cells were stimulated with LPS (1 μg/mL) and incubated in the presence or absence of increasing concentrations of GTEO (25–200 µg/mL) for 1 h. The phosphorylated and total levels of I-κBα and IKKα were determined by immunoblotting with specific antibodies. GAPDH served as an internal loading control. Histogram shows the relative protein expression levels of iNOS and COX-2, which are normalized with an internal control GAPDH. Data are reported as mean ± SD of three independent experiments. ^Δ^
*p* < 0.001 indicates a significant difference between the control and LPS-only treated groups. ** *p* < 0.01, and *** *p* < 0.001 show significant differences between the LPS-alone and GTEO treatment groups.

**Figure 5 plants-12-01241-f005:**
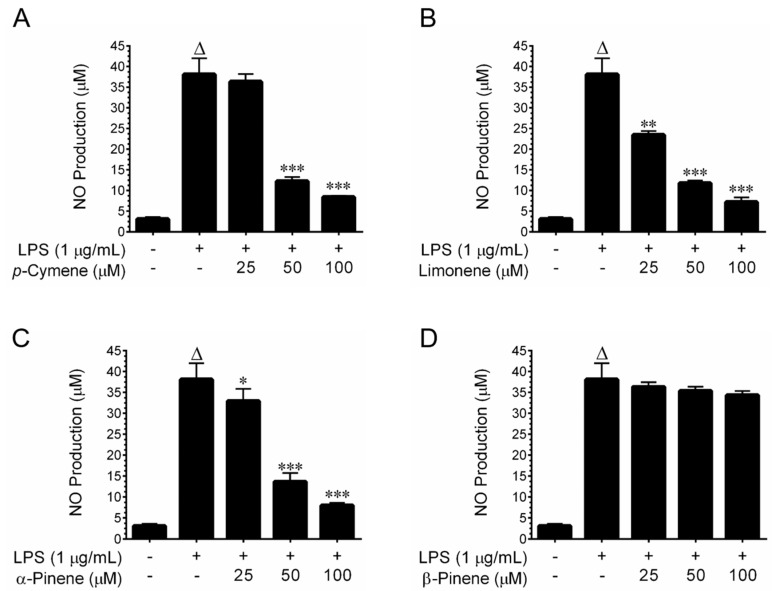
Effect of *p*-cymene, α-pinene, β-pinene, and limonene on LPS-induced NO production. RAW 264.7 cells were stimulated with LPS (1 μg/mL) and incubated with the presence or absence of increasing concentrations of test compounds (25–100 µM) and in the presence of 1 μg/mL LPS for 24 h. (**A**–**D**) The nitrite concentration in the culture media was determined by the Griess reagent assay. Data are reported as mean ± SD of three independent experiments. ^Δ^
*p* < 0.001 indicates a significant difference between the control and LPS-only treated groups. * *p* < 0.05, ** *p* < 0.01, and *** *p* < 0.001 show significant differences between the LPS-alone and GTEO treatment groups.

**Figure 6 plants-12-01241-f006:**
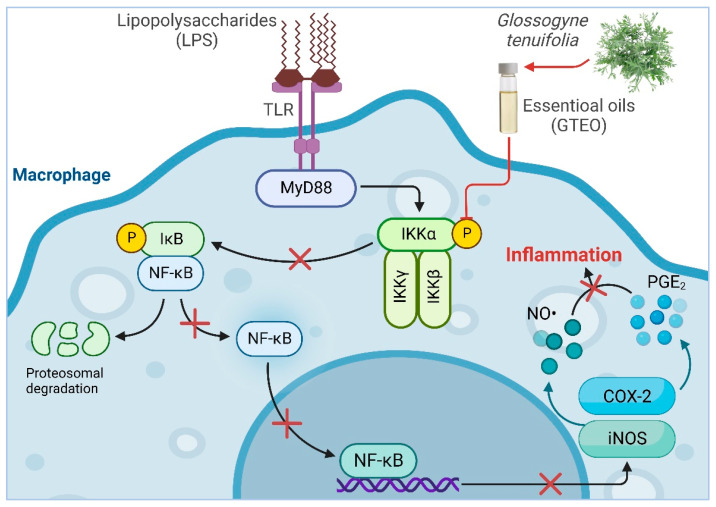
The schematic representation illustrates GTEO’s anti-inflammatory properties.

**Table 1 plants-12-01241-t001:** The major components and their relative contents (%) of *G. tenuifolia* essential oil.

RT (min)	Compounds	Concentration (%)	KI ^a^	Identification ^b^
10.29	α-Pinene	6.69	935	KI/MS
12	β-Pinene	3.16	975	KI/MS
12.19	β-Myrcene	14.68	979	KI/MS
14.36	*p*-Cymene	35.3	1026	KI/MS
14.56	_D_-Limonene	5.17	1031	KI/MS
14.63	*cis*-β-Ocimene	8.49	1032	KI/MS
15.93	*trans*-Sabinene hydrate	0.31	1060	KI/MS
17.95	Linalool	0.82	1100	KI/MS
21.74	Terpinen-4-ol	1.55	1182	KI/MS
22.03	2-Cyclohexen-1-one, 4-(1-methylethyl)	1.14	1187	KI/MS
22.44	α-Terpineol	0.89	1195	KI/MS
24.61	Benzaldehyde, 4-(1-methylethyl)-	1.05	1244	KI/MS
26.21	Phellandral	2.9	1278	KI/MS
27.01	Sabinyl acetate	0.2	1295	KI/MS
27.19	Carvacrol	0.38	1298	KI/MS
28.24	Myrtenyl acetate	0.55	1323	KI/MS
30.66	(*E*)-β-Damascenone	0.26	1378	KI/MS
32.3	β-Cedrene	9.8	1416	KI/MS
35.05	Germacrene D	1.02	1483	KI/MS
38.16	Nerolidol	0.36	1561	KI/MS
38.93	Caryophyllene oxide	1.23	1580	KI/MS

^a^ Kovats index on DB-5MS column in reference to *n*-alkanes. ^b^ MS, NIST library and literature; KI, Kovats index.

## Data Availability

The data presented in this study are available on request from the corresponding author.

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
