# Peer review of "Essential Oil from Glossogyne tenuifolia Inhibits Lipopolysaccharide-Induced Inflammation-Associated Genes in Macro-Phage Cells via Suppression of NF-κB Signaling Pathway"

_plants, 2023, doi:10.3390/plants12061241_

Round 1

Reviewer 1 Report

 The manuscript entitled “Essential Oil from Glossogyne tenuifolia Inhibits Lipopolysaccharide-Induced Inflammation-Associated Genes in Macro phage Cells via Suppression of NF-kB Signaling Pathway” by Lin et al discusses the anti-inflammatory properties of G. tenuifolia essential oil (GTEO) on lipopolysaccharide (LPS)-induced inflammation in murine macrophage cells (RAW 264.7) in vitro. Overall, the study is clear and concise. The introduction is relevant and theory-based. Sufficient information about the present study rationale and procedures is provided for the readers. The methods are generally appropriate, although clarification of a few details is required. 

The manuscript requires a major revision before it can get published in Plants.

Specific Comments below:

  1. I am happy to see that the GTEO can neutralize a very high concentration of LPS (1 ug/mL). But the authors never discussed the mechanism of action of the oil, let alone measure it. Does it directly bind to the LPS to show its anti-inflammatory activity? Or does it enter the cell and act on IKK alpha as Figure 7 suggests?
  2. MTT assay needs to be performed on 3T3 cells to truly evaluate the cytotoxicity profile of GTEO.
  3. The GAPDH which is the loading control for western blot experiments seems to be not equal. Please add the densitometry data for the western blot assays.
  4. NO estimation is reported in % of control. That is not Nitrite production is reported in Griess Reagent. Please report it in a micro molar.
  5. The Materials and Methods section needs the attention of the authors. Please refer to the manuscript “Selective phenylalanine to proline substitution for improved antimicrobial and anticancer activities of peptides designed on phenylalanine heptad repeat" by Tripathi et al to take help in methods revision. They have done similar experiments as yours”. 
  6. The oil extraction process needs further details. Section 4.1
  7. I do not fully agree with the cell viability formula in Section 4.3. The absorbance, which was proportional to cell viability, should be measured at 570 nm and a reference wavelength of 630 nm should be taken as background. Please subtract the background from the OD obtained at 570 during calculations.
  8. Please provide the GC-MS data as supplementary.
  9. Line 373: Did you use 10 M of working primers? Or is that a typo error?
  10. There is no mention of the PCR program that was used to run the PCR for these genes. How would the readers be able to execute the PCR experiments? Also, mention the method that was used to quantify the PCR data. I believe it was a double delta method.

Author Response

Response to Reviewer’s Comments

 Reviewer: 1

Reviewer’s opinion: The manuscript entitled “Essential Oil from Glossogyne tenuifolia Inhibits Lipopolysaccharide-Induced Inflammation-Associated Genes in Macro phage Cells via Suppression of NF-kB Signaling Pathway” by Lin et al discusses the anti-inflammatory properties of G. tenuifolia essential oil (GTEO) on lipopolysaccharide (LPS)-induced inflammation in murine macrophage cells (RAW 264.7) in vitro. Overall, the study is clear and concise. The introduction is relevant and theory-based. Sufficient information about the present study rationale and procedures is provided for the readers. The methods are generally appropriate, although clarification of a few details is required. The manuscript requires a major revision before it can get published in Plants.

Response:  We would appreciate reviewer-1, who provided a positive comment on our work. The reviewer’s comments and our responses are as follows.

Comment. 1: I am happy to see that the GTEO can neutralize a very high concentration of LPS (1 ug/mL). But the authors never discussed the mechanism of action of the oil, let alone measure it. Does it directly bind to the LPS to show its anti-inflammatory activity? Or does it enter the cell and act on IKK alpha as Figure 7 suggests?

Response: Thank you for your query. Based on our data, we conclude that essential oil enters into cells and acts as an IKKa repressor, as evidenced by inhibiting LPS-induced phosphorylation of IKKa (Figure 4).

Comment. 2: MTT assay needs to be performed on 3T3 cells to truly evaluate the cytotoxicity profile of GTEO.

Response:  Thank you for your suggestion. But, we apologize that we could not perform this experiment within the revising time period, because the 3T3 cell line is currently not available in our laboratory. However, we seriously consider your suggestion. In the nearing future, we will evaluate the cytotoxic effect of essential oils in 3T3 cells.

Comment. 3: The GAPDH which is the loading control for western blot experiments seems to be not equal. Please add the densitometry data for the western blot assays.

Response:  Thank you for your suggestion. Actually, in all western blots, the protein of interest was normalized with GAPDH and the corresponding densitometry data can be found in the histogram, which is supplied to the side of the western blots (Figure 2C & Figure 4). In addition, we included this information in the figure legends (Lines 129-130; 174-175).

Comment. 4: NO estimation is reported in % of control. That is not Nitrite production reported in Griess Reagent. Please report it in a micromolar.

Response:  Thank you for pointing this out. As per your suggestion, the NO production was calculated in a micromolar concentration and re-plotted in Figure 1B and Figure 5.

Comment. 5: The Materials and Methods section needs the attention of the authors. Please refer to the manuscript “Selective phenylalanine to proline substitution for improved antimicrobial and anticancer activities of peptides designed on phenylalanine heptad repeat" by Tripathi et al to take help in methods revision. They have done similar experiments as yours”.

Response: Dear reviewer, we believe this comment is inappropriate for our work.

Comment. 6: The oil extraction process needs further details. Section 4.1.

Response: Thank you. As per your suggestion, we included detailed information for essential oil extraction (Lines 309-313).

Comment. 7: I do not fully agree with the cell viability formula in Section 4.3. The absorbance, which was proportional to cell viability, should be measured at 570 nm and a reference wavelength of 630 nm should be taken as background. Please subtract the background from the OD obtained at 570 during calculations.

Response: Thank you for pointing this out. As per your suggestion, we revised the calculation of the percentage of cell viability as stated in Lines 337-339.

Comment. 8: Please provide the GC-MS data as supplementary.  

Response: Thank you.  As per your suggestion, the GC-MS data was provided in the supplementary information (Figure S1).

Comment. 9: Line 373: Did you use 10 M of working primers? Or is that a typo error?

Response: Thank you for pointing this out. It is a typo error. We corrected this in the revised manuscript (Line 359).

Comment. 10: There is no mention of the PCR program that was used to run the PCR for these genes. How would the readers be able to execute the PCR experiments? Also, mention the method that was used to quantify the PCR data. I believe it was a double delta method.

Response: Thank you for pointing this out. As per your suggestion, we included the PCR program that was used to run these genes (Lines 360-361) and the quantification of PCR products was described in Lines 368-370.

Reviewer 2 Report

Figures 1 and 2 are in the Introduction, they belong to the results.

This problem extends throughout the entire work. For example Figures 3 are listed in 2.3. GTEO Inhibits LPS-Induced iNOS and COX-2 Expression in RAW 264.7 Cells, cited in 2.4

In Table 1, what does concentration mean and how is it calculated, why are the percentages in brackets

The components in Table 1 and in the main text are not written correctly, pay attention to the writing in italics

In the identification of compounds in the composition of the essential oil, the literature was not used: Adams, R.P. Identification of Essential Oil Components by Gas Chromatography/Mass Spectrometry, 4th ed.; Allured Publishing: Carol Stream, IL, USA, 2017; ISBN 978-1-932633-21-4.

In results 2.4. GTEO Suppressed LPS-Induced NF-B Transcriptional Activity in RAW 264.7 Cells cited reference 20. Why are the results presented here

Also the results of 2.7. Nitric Oxide Inhibitory Effects of Major Constituents of GTEO, cited in reference 19.

Why is Figure 7 not cited anywhere in materials and methods. This Figure could be in the Introduction.

Figure 6 is mentioned in the Conclusion, why it was not mentioned anywhere before.

Author Response

Response to Reviewer’s Comments

Reviewer: 2

Comment. 1: Figures 1 and 2 are in the Introduction, they belong to the results. This problem extends throughout the entire work. For example Figures 3 are listed in 2.3. GTEO Inhibits LPS-Induced iNOS and COX-2 Expression in RAW 264.7 Cells, cited in 2.4.

Response: Thank you for pointing this out. As per your suggestion, we reorganized all the figures into an appropriate place.

Comment. 2: In Table 1, what does concentration mean and how is it calculated, why are the percentages in brackets?

Response:  Thank you for your query. The concentration in percentage indicates, the percentage of individual compounds in essential oil. Usually, the area below the GC-MS absorption peak is calculated based on the mathematical relations and the programming algorithm of the device software, and the relevant value can be presented as the concentration of individual compounds.

Comment. 3: The components in Table 1 and in the main text are not written correctly, pay attention to the writing in italics.

Response:  Thank you for pointing this out. We have corrected this in our revised manuscript.

Comment. 4: In the identification of compounds in the composition of the essential oil, the literature was not used: Adams, R.P. Identification of Essential Oil Components by Gas Chromatography/Mass Spectrometry, 4th ed.; Allured Publishing: Carol Stream, IL, USA, 2017; ISBN 978-1-932633-21-4.

Response:  Thank you for pointing this out. As per your suggestion, we included this reference in our revised manuscript (Lines 419; 538-539).

Comment. 5: In results 2.4. GTEO Suppressed LPS-Induced NF-kB Transcriptional Activity in RAW 264.7 Cells cited reference 20. Why are the results presented here? Also the results of 2.7. Nitric Oxide Inhibitory Effects of Major Constituents of GTEO, cited in reference 19.

Response: Thank you for your query. Liu et al., 2017 [ref # 20] described that “NF-kB is a transcription factor t regulating pro-inflammatory genes including iNOS, COX-2, TNF-a, and IL-1b”. Therefore, we cited reference # 20 after the statement.

In a similar way, Chyau et al., 2007 [ref # 19] demonstrated that “a-pinene, b-pinene, limonene, p-cymene, and b-phellandrene are major components of GTEO” and we cited reference # 19 after the statement.

Comment. 6: Why is Figure 7 not cited anywhere in materials and methods? This Figure could be in the Introduction.

Response: Thank you for pointing this out. It is a typo error. We wrongly named Figure 6 as Figure 7 (Line 442).

Comment. 7: Figure 6 is mentioned in the Conclusion, why it was not mentioned anywhere before.

Response: Thank you for your query. Figure 6 is the overall conclusion of our work. That is why we placed Figure 6 in the conclusion section.

Reviewer 3 Report

The manuscript “Essential Oil from Glossogyne tenuifolia Inhibits Lipopolysaccharide-Induced Inflammation-Associated Genes in Macrophage Cells via Suppression of NF-κB Signaling Pathwayprovides novel and scientifically relevant information about immunomodulatory potential of Glossogyne tenuifoila essential oil. The research is well organized, and manuscript is well written.

Please correct some minor issues:

Line 52-53 - please rephrase the sentence, it is not understandable. Instead of “debilitating” I suggest using “chronic” or “chronic inflammatory”

Line 105 – “were” is used twice in the same sentence

Line 69-96 and 111-121 - I would suggest that the figures be placed within the “Results” section

Line 111 – in Figure 2A and 2B, under the second column is stated that RAW 264.7 cells were treated with LPS and GTEO.  I suppose that cells were treated only with LPS, without GTEO, for the purpose of positive control. The figures are labeled in capital letters (1A, 1B, 2A, 2B) so I suggest keeping the labels consistent in the text (not 1a, 1b …). Please correct further in the manuscript.

Line 199 – Please explain why you have not identified more compounds in essential oil? In line 298 you cited that essential oil has 62 compounds, but also only 30 were identified. I suggest that you list all the isolated peaks in the Table 1, even though they were not identified.

Line 215 – please correct  in “…β-mircene exibitetd cytotoxicity…”

Lines 318-324 – How are γ-muurolene, β-phellandrene and cryptone relevant for this study? These constituens were not found in GTEO isolated in this study. In line 321 there is a sentence “Therefore, we examined the NO inhibitory effect of muurolene“. I do not find these results in the manuscript (not in Results, nor in Figures)

Line 331 – please check that all the used chemicals are properly listed in Chemicals and reagents part. For example, GAPDH is missing the full name.

Line 325 – Materials and methods – Have you used previously described protocols for 4.1; 4.5; 4.6; 4.7? If yes please provide the references.

Line 440 – Figure 6 instead Figure 7

Author Response

Response to Reviewer’s Comments

Reviewer: 3

Reviewer’s opinion: The manuscript “Essential Oil from Glossogyne tenuifolia Inhibits Lipopolysaccharide-Induced Inflammation-Associated Genes in Macrophage Cells via Suppression of NF-κB Signaling Pathway“ provides novel and scientifically relevant information about immunomodulatory potential of Glossogyne tenuifoila essential oil. The research is well organized, and manuscript is well written.

Response:  We would appreciate reviewer-3, who provided a positive comment on our work. The reviewer’s comments and our responses are as follows.

Comment. 1: Line 52-53 - please rephrase the sentence, it is not understandable. Instead of “debilitating” I suggest using “chronic” or “chronic inflammatory”.

Response: Thank you for pointing this out. As per your suggestion, we rephrased the sentence (Line 51).

Comment. 2: Line 105 – “were” is used twice in the same sentence.

Response:  Thank you for pointing this out. As per your suggestion, we rephrased the sentence (Line 76).

Comment. 3: Line 69-96 and 111-121 - I would suggest that the figures be placed within the “Results” section.

Response:  Thank you for pointing this out. As per your suggestion, we reorganized all the figures into an appropriate place.

Comment. 4: Line 111 – in Figure 2A and 2B, under the second column is stated that RAW 264.7 cells were treated with LPS and GTEO.  I suppose that cells were treated only with LPS, without GTEO, for the purpose of positive control. The figures are labeled in capital letters (1A, 1B, 2A, 2B) so I suggest keeping the labels consistent in the text (not 1a, 1b …). Please correct further in the manuscript.

Response:  Thank you for pointing this out. As per your suggestion, we rephrased the sentences (Lines 101-103; 122-124; 152-154; 170-172; 208-209). Also, in our revised version, we replaced figure labels with capital letters.

Comment. 5: Line 199 – Please explain why you have not identified more compounds in essential oil? In line 298 you cited that essential oil has 62 compounds, but also only 30 were identified. I suggest that you list all the isolated peaks in Table 1, even though they were not identified.

Response: Thank you for your query. Based on our extraction method (hydro-distillation), we could find only 21 compounds in our sample. All 21 compounds were chemically identified. Whereas, a previous study by Chyau et al., 2007 used the steam distillation and solvent extraction (SDE) method to obtain the essential oil. Due to the variation in experimental methods, the number of chemical constituents greatly varies.

Comment. 6: Line 215 – please correct  in “…β-mircene exibitetd cytotoxicity…”

Response: Thank you for pointing this out. In our revised version, we rephrased the sentence (Line 199).

Comment. 7: Lines 318-324 – How are γ-muurolene, β-phellandrene and cryptone relevant for this study? These constituens were not found in GTEO isolated in this study. In line 321 there is a sentence “Therefore, we examined the NO inhibitory effect of muurolene“. I do not find these results in the manuscript (not in Results, nor in Figures).

Response: Thank you for pointing this out. We apologize that we wrongly phrased this information. In our revised manuscript, we rephrased this sentence (Lines 303-306).

Comment. 8: Line 331 – please check that all the used chemicals are properly listed in Chemicals and reagents part. For example, GAPDH is missing the full name..

Response: Thank you for pointing this out. As per your suggestion, we included the abbreviation for GAPDH (Lines 324).

Comment. 9: Line 325 – Materials and methods – Have you used previously described protocols for 4.1; 4.5; 4.6; 4.7? If yes please provide the references.

Response: Thank you for your suggestion. These protocols are not followed by anyone. Therefore, we believe it is not appropriate to cite the references. 

Comment. 10: Line 440 – Figure 6 instead of Figure 7.

Response: Thank you for pointing this out. As per your suggestion, we corrected this typo (Lines 442).

Round 2

Reviewer 1 Report

I AM STILL NOT SURE THAT THE WORKING STOCK OF PRIMERS  USED WAS 10 nM. LINE 359.

USALLY 10 MICROMOLAR OF WORKING PRIMER CONCENTRATION IS USED FOR PCR REACTIONS

Author Response

Minor Comment. 1: I AM STILL NOT SURE THAT THE WORKING STOCK OF PRIMERS USED WAS 10 nM. LINE 359. USUALLY 10 MICROMOLAR OF WORKING PRIMER CONCENTRATION IS USED FOR PCR REACTIONS.

Response: Thank you once again for correcting us. Yes, the primer concentration is 10 micromolar. We corrected this typo in our revised manuscript (Line 360).

Reviewer 2 Report

Please fix: the components in Table 1 and in the main text are not written correctly, pay attention to the writing in italics.

Author Response

Minor Comment. 1: Please fix: the components in Table 1 and in the main text are not written correctly, pay attention to the writing in italics.

Response: Thank you once again for correcting us. We have corrected these typos in our revised manuscript (Table. 1) and Line 552.